# RETHINKING LEARNING RATE SCHEDULES FOR STOCHASTIC OPTIMIZATION

## ABSTRACT

There is a stark disparity between the learning rate schedules used in the practice of large scale machine learning and what are considered admissible learning rate schedules prescribed in the theory of stochastic approximation. Recent results, such as in the 'super-convergence' methods which use oscillating learning rates, serve to emphasize this point even more. One plausible explanation is that non-convex neural network training procedures are better suited to the use of fundamentally different learning rate schedules, such as the "cut the learning rate every constant number of epochs" method (which more closely resembles an exponentially decaying learning rate schedule); note that this widely used schedule is in stark contrast to the polynomial decay schemes prescribed in the stochastic approximation literature, which are indeed shown to be (worst case) optimal for classes of convex optimization problems.

The main contribution of this work shows that the picture is far more nuanced, where we do not even need to move to non-convex optimization to show other learning rate schemes can be far more effective. In fact, even for the simple case of stochastic linear regression with a fixed time horizon, the rate achieved by any polynomial decay scheme is suboptimal compared to the statistical minimax rate (by a factor of condition number); in contrast the "cut the learning rate every constant number of epochs" provides an exponential improvement (depending only logarithmically on the condition number) compared to any polynomial decay scheme. Finally, it is important to ask if our theoretical insights are somehow fundamentally tied to quadratic loss minimization (where we have circumvented minimax lower bounds for more general convex optimization problems)? Here, we conjecture that recent results which make the gradient norm small at a near optimal rate, for both convex and non-convex optimization, may also provide more insights into learning rate schedules used in practice.

## 1 INTRODUCTION

The recent advances in machine learning and deep learning rely almost exclusively on stochastic optimization methods, primarily SGD and its variants. Here, these large scale stochastic optimization methods are manually (and often painstakingly) tuned to the problem at hand (often with parallelized hyper-parameter searches), where there is, as of yet, no class of "universal methods" which uniformly work well on a wide range of problems with little to no hyper-parameter tuning. This is in stark contrast to non-stochastic numerical optimization methods, where it is not an overstatement to argue that the l-BFGS and non-linear conjugate gradient methods (with no hyper-parameter tuning whatsoever) have provided nearly unbeatable procedures (for a number of decades) on nearly every unconstrained convex and non-convex problem. In the land of stochastic optimization, there are two dominant (and somewhat compatible approaches): those methods which often manually tune learning rate schedules to achieve the best performance (Krizhevsky et al., 2012; Sutskever et al., 2013; Kingma & Ba, 2014; Kidambi et al., 2018) and those methods which rely on various forms of approximate preconditioning (Duchi et al., 2011; Tieleman & Hinton, 2012; Kingma & Ba, 2014). This works examines the former class of methods, where we seek a more refined understanding of the issues of learning rate scheduling, through both theoretical analysis and empirical studies.

Learning rate schedules for SGD is a rather enigmatic topic since there is a stark disparity between what is considered admissible in theory and what is employed in practice to achieve the best re-

sults. Let us elaborate on this distinction more clearly. In theory, a vast majority of works starting with Robbins & Monro (1951); Polyak & Juditsky (1992) consider learning rates that have the form of $\eta_t = \frac{a}{b+t^\alpha}$ for some $a, b \geq 0$ and $1/2 < \alpha \leq 1$ – we call these polynomial decay schemes. The key property enjoyed by these polynomial decay schemes is that they are not summable but are square summable. A number of works obtain bounds on the asymptotic convergence rates of such schemes. Note that the focus of these works is to design learning rate schemes that work well for *all* large values of $t$. In contrast, practitioners are interested in achieving the best performance given a computational budget or equivalently a fixed time horizon $T$ e.g., 100 passes on training dataset with a batch size of 128.

The corresponding practically best performing learning rate scheme is often one where the step size is cut by a constant factor once every few epochs, or, equivalently, when no progress is made on a validation set (Krizhevsky et al., 2012; He et al., 2016b) (often called a dev set based decay scheme). Such schemes are widely popular to the extent that they are available as schemes in deep learning libraries such as PyTorch [1] and several such useful tools of the trade are taught on popular deep learning courses [2]. Furthermore, what is (often) puzzling (from a theory perspective) is the emphasis that is laid on "babysitting" the learning rates [3] to achieve the best performance. Why do practitioners use constant and cut learning rate schemes while most of the theory work routinely works with polynomial decaying schemes? Of course, implicit to this question is the view that both of these schemes are not equivalent. Indeed if both of these were equivalent, one could parameterize the learning rate as $\frac{a}{b+t^\alpha}$ and do hyperparameter search over $a, b$ and $\alpha$. In practice, this simply does not give results comparable to the constant and cut schemes.[4] One potential explanation for this could be that, in the context of neural network training, local minima found by constant and cut schemes are of much better quality than those found by polynomial decay schemes, while for convex problems, polynomial decay schemes are indeed optimal.

The primary contribution of this work is to show that this is simply not the case. We concretely show how minimax optimal theoretical learning rates (i.e. polynomial decay schemes for wide classes of convex optimization problems) may be misleading (and sub-optimal for locally quadratic problems), and the story in practice is more nuanced. There important issues at play with regards to this suboptimality. First, even for the simple case of stochastic linear regression, with a fixed time horizon, the rate achieved by any polynomial decay scheme (i.e., any choice of $a, b$ and $\alpha$) is suboptimal compared to the statistical minimax rate (i.e., information theoretically best possible rate achievable by any algorithm) by a factor of condition number $\kappa$ (see Section 3 for definitions), while there exist constant and cut schemes that are suboptimal only by a factor of $\log \kappa$.

Second, this work shows that a factor of $\kappa$ suboptimality is unavoidable if we wish to bound the error of *each* iterate of SGD. In other words, we show that the convergence rate of $\limsup$ of the error, as $t \to \infty$, has to be necessarily suboptimal by a factor of $\tilde{\Omega}(\kappa)$ compared to the statistical minimax rate, for *any* learning rate sequence (polynomial or not). In fact, at least $\tilde{\Omega}1/\kappa$ fraction of the iterates have this suboptimality. With this result, things become quite clear – all the works in stochastic approximation try to bound the error of *each* iterate of SGD asymptotically (or $\limsup$ of the error in other words). Since this necessarily has to be suboptimal by a factor of $\tilde{\Omega}(\kappa)$ compared to the statistical minimax rates, the suboptimality of polynomial decay rates is not an issue. However, with a fixed time horizon, there exist learning rate schemes with much better convergence rates, while polynomial decay schemes fail to get better rates in this simpler setting (of known time horizon).

Thirdly, the work shows that, for stochastic linear regression, if we consider $\liminf$ (rather than $\limsup$) of the error, it is possible to design schemes that are suboptimal by only a factor of $\log \kappa$ compared to the minimax rates. Variants of the constant and cut schemes achieve this guarantee.

In summary, the contributions of this paper are showing how widely used pratical learning rate schedules are, in fact, highly effective even in the convex case. In particular, our theory and empirical results demonstrate this showing that:

---

[1] https://pytorch.org/docs/stable/optim.html#torch.optim.lr_scheduler.ReduceLROnPlateau

[2] http://cs231n.github.io/

[3] http://cs231n.github.io/neural-networks-3/

[4] In fact, this work shows an instance where there is a significant (provable) difference between the performance of these two schemes.

- For a fixed time horizon, constant and cut schemes are provably, significantly better than polynomial decay schemes.
- There is a fundamental difference between fixed time horizon and infinite time horizon.
- The above difference can be mitigated by considering $\liminf$ of error instead of $\limsup$.
- In addition to our theoretical contributions, we empirically verify the above claims for neural network training on cifar-10.

Extending results on the performance of constant and cut schemes to more general convex optimization problems, beyond stochastic linear regression, is an important future direction. However, the fact that the suboptimality of polynomial decay schemes even for the simple case of stochastic linear regression, has not been realized after decades of research on stochastic approximation is striking.

In summary, the results of this paper show that, even for stochastic linear regression, the popular in practice, constant and cut learning rate schedules are provably better than polynomial decay schemes popular in theory and that there is a need to rethink learning rate schemes and convergence guarantees for stochastic approximation. Our results also suggest that current approaches to hyperparameter tuning of learning rate schedules might not be right headed and further suggest potential ways of improving them.

**Paper organization**: The paper is organized as follows. We review related work in Section 2. Section 3 describes the notation and problem setup. Section 4 presents our results on the suboptimality of both polynomial decay schemes and constant and cut schemes. Section 5 presents results on infinite horizon setting. Section 6 presents experimental results and Section 7 concludes the paper.

## 2 RELATED WORK

We will split related work into two parts, one based on theory and the other based on practice.

**Related efforts in theory:** SGD and the problem of stochastic approximation was introduced in the seminal work of Robbins & Monro (1951); this work also elaborates on stepsize schemes that are satisfied by asymptotically convergent stochastic gradient methods: we refer to these schemes as "convergent" stepsize sequences. The (asymptotic) statistical optimality of iterate averaged SGD with larger stepsize schemes of $O(1/n^\alpha)$ with $\alpha \in (0.5, 1)$ was proven in the seminal works of Ruppert (1988); Polyak & Juditsky (1992). The notions of convergent learning rate schemes in stochastic approximation literature has been studied in great detail (Ljung et al., 1992; Kushner & Yin, 2003; Bharath & Borkar, 1999; Lai, 2003). Nearly all of the aforementioned works rely on function value sub-optimality to measure convergence and rely on the notion of asymptotic convergence (i.e. in the limit of the number of updates of SGD tending to infinity) to derive related "convergent stepsize schedules". Along this line of thought, there are several efforts that prove (minimax) optimality of the aforementioned rates (in a worst case sense and not per problem sense) e.g., Nemirovsky & Yudin (1983); Raginsky & Rakhlin (2011); Agarwal et al. (2012).

An alternative viewpoint is to consider gradient norm as a means to measure the progress of an algorithm. Along this line of thought are several works including the stochastic process viewpoint considered by Polyak & Juditsky (1992) and more recently, the work of Nesterov (2012) (working with deterministic (exact) gradients). The work of Allen-Zhu (2018) considers questions relating to making the gradient norm small when working with stochastic gradients, and provides an improved rate. We return to this criterion in Section 7.

In terms of oracle models, note that both this paper, as well as other results (Lacoste-Julien et al., 2012; Rakhlin et al., 2012; Bubeck, 2014), work in an oracle model that assumes bounded variance of stochastic gradients or similar assumptions. There is an alternative oracle model for analyzing SGD as followed in papers includingBach & Moulines (2013); Bach (2014); Jain et al. (2017) which is arguably more reflective of SGD's behavior in practice. For more details, refer to Jain et al. (2017). It is an important direction to prove the results of this paper working in the alternative practically more applicable oracle model.

**Efforts in practice:** As highlighted in the introduction, practical efforts in stochastic optimization have diverged from the classical theory of stochastic approximation, with several deep learning

libraries like pytorch [5] providing unconventional alternatives such as cosine/sawtooth/dev set decay schemes, or even exponentially decaying learning rate schemes. In fact, a natural scheme used in training convolutional neural networks for vision is where the learning rate is cut by a constant factor after a certain number of epochs. Such schemes are essentially discretized variants of exponentially decaying learning rate schedules. We note that there are other learning rate schedules that have been recently proposed such as sgd with warm restarts (Loshchilov & Hutter, 2016), oscillating learning rates (Smith & Topin, 2017) etc., that are unconventional and have attracted a fair bit of attention. Furthermore, exponential learning rates appear to be considered in more recent NLP papers (see for e.g., Krishnamurthy et al. (2017)) [6].

## 3 PROBLEM SETUP

**Notation**: We represent scalars with normal font $a, b, L$ etc., vectors with boldface lowercase characters $\mathbf{a}, \mathbf{b}$ etc. and matrices with boldface uppercase characters $\mathbf{A}, \mathbf{B}$ etc. We represent positive semidefinite (PSD) ordering between two matrices using $\succeq$. The symbol $\gtrsim$ represents that the direction of inequality holds for some universal constant.

Our theoretical results focus on the following additive noise stochastic linear regression problem. We present the setup and associated notation in this section. We wish to solve:
$$\min_{\mathbf{w} \in \mathbb{R}^d} f(\mathbf{w}) \text{ where } f(\mathbf{w}) \overset{\text{def}}{=} \frac{1}{2}\mathbf{w}^\top \mathbf{H}\mathbf{w} - \mathbf{w}^\top \mathbf{b}$$

for some positive definite matrix $\mathbf{H}$ and vector $\mathbf{b}$.[7] We denote the smallest and largest eigenvalues of $\mathbf{H}$ by $\mu > 0$ and $L > 0$. $\kappa \overset{\text{def}}{=} \frac{L}{\mu}$ denotes the condition number of $\mathbf{H}$. We have access to a stochastic gradient oracle which gives us $\widehat{\nabla} f(\mathbf{w}) = \nabla f(w) + \mathbf{e}$, where $\mathbf{e}$ is a random vector satisfying[8] $\mathbb{E}[\mathbf{e}] = 0$ and $\mathbb{E}[\mathbf{e}\mathbf{e}^\top] = \sigma^2 \mathbf{H}$.

Given an initial point $\mathbf{w}_0$ and step size sequence $\eta_t$, the SGD algorithm proceeds with the update
$$\mathbf{w}_t = \mathbf{w}_{t-1} - \eta_t \widehat{\nabla} f(\mathbf{w}_{t-1}) = \mathbf{w}_{t-1} - \eta_t \left(\nabla f(\mathbf{w}_{t-1}) + \mathbf{e}_t\right),$$
where $\mathbf{e}_t$ are independent for various $t$ and satisfy the above mean and variance conditions.

Let $\mathbf{w}^* \overset{\text{def}}{=} \arg\min_{\mathbf{w} \in \mathbb{R}^d} f(\mathbf{w})$. The suboptimality of a point $\mathbf{w}$ is given by $f(\mathbf{w}) - f(\mathbf{w}^*)$. It is well known that given $t$ accesses to the stochastic gradient oracle above, any algorithm that uses these stochastic gradients and outputs $\widehat{\mathbf{w}}_t$ has suboptimality that is lower bounded by $\frac{\sigma^2 d}{t}$. More concretely (Van der Vaart, 2000), we have that
$$\lim_{t \to \infty} \frac{\mathbb{E}[f(\widehat{\mathbf{w}}_t)] - f(\mathbf{w}^*)}{\sigma^2 d/t} \geq 1.$$

Moreover there exist schemes that achieve this rate of $(1 + o(1))\frac{\sigma^2 d}{t}$ e.g., constant step size SGD with averaging (Polyak & Juditsky, 1992). This rate of $\sigma^2 d/t$ is called the statistical minimax rate.

## 4 COMPARISON BETWEEN POLYNOMIAL DECAY SCHEMES VS CONSTANT AND CUT SCHEMES

In this section, we will show that polynomial decay schemes are suboptimal compared to the statistical minimax rate by at least a factor of $\kappa$ while constant and cut schemes are suboptimal by at most a factor of $\log \kappa$.

---

[5]see https://pytorch.org/docs/stable/optim.html for a complete list of alternatives

[6]Refer to their JSON file https://github.com/allenai/allennlp/blob/master/training_config/wikitables_parser.jsonnet

[7]Any linear least squares $\frac{1}{2n}\sum_{i=1}^{n}\left(\mathbf{x}_i^\top \mathbf{w} - y_i\right)^2$ can be written as above with $\mathbf{H} \overset{\text{def}}{=} \frac{1}{n}\sum_i \mathbf{x}_i \mathbf{x}_i^\top$ and $\mathbf{b} \overset{\text{def}}{=} \frac{1}{n}\sum_i y_i \mathbf{x}_i$.

[8]While this might seem very special, this is indeed a fairly natural scenario. For instance, in stochastic linear regression with independent additive noise, i.e., $y_t = \mathbf{x}_t^\top \mathbf{w}^* + \epsilon_t$ where $\epsilon_t$ is a random variable independent of $\mathbf{x}_t$ and $\mathbb{E}[\epsilon_t] = 0$ and $\mathbb{E}[\epsilon_t^2] = \sigma^2$, the noise in the gradient has this property. On the other hand, the results in this paper can also be generalized to the setting where $\mathbb{E}[\mathbf{e}\mathbf{e}^\top] = \mathbf{V}$ for some arbitrary matrix $\mathbf{V}$. However, error covariance of $\sigma^2 \mathbf{H}$ significantly simplifies exposition.

## 4.1 SUBOPTIMALITY OF POLYNOMIAL DECAY SCHEMES

Our first result shows that there exist problem instances where all polynomial decay schemes i.e., those of the form $\frac{a}{b+t^\alpha}$, for any choice of $a, b$ and $\alpha$ are suboptimal by at least a factor of $\Omega(\kappa)$ compared to the statistical minimax rate.

**Theorem 1.** *There exists a problem instance such that the initial function value $f(\mathbf{w}_0) \leq \sigma^2 d$, and for any fixed time $T$ satisfying $T \geq \kappa^2$, for all $a, b \geq 0$ and $0.5 \leq \alpha \leq 1$, and for the learning rate scheme $\eta_t = \frac{a}{b+t^\alpha}$, we have $\mathbb{E}\left[f(\mathbf{w}_T)\right] - f(\mathbf{w}^*) \geq \frac{\kappa}{32} \cdot \frac{\sigma^2 d}{T}$.*

## 4.2 SUBOPTIMALITY OF CONSTANT AND CUT SCHEME

Our next result shows that there exist constant and cut schemes that achieve statistical minimax rate upto a multiplicative factor of only $\log \kappa \log^2 T$.

**Theorem 2.** *For any problem and fixed time horizon $T > \kappa \log(\kappa)$, there exists a constant and cut learning rate scheme that achieves $\mathbb{E}\left[f(\mathbf{w}_T)\right] - f(\mathbf{w}^*) \leq \frac{f(\mathbf{w}_0) - f(\mathbf{w}^*)}{T^3} + 2 \log \kappa \cdot \log^2 T \cdot \frac{\sigma^2 d}{T}$.*

We will now consider an exponential decay scheme (in contrast to polynomial ones from Section 4.1) which is a smoother version of constant and cut scheme. We show that the same result above for constant and cut scheme can also be extended to the exponential decay scheme.

**Theorem 3.** *For any problem and fixed horizon $T$, there exist constants $a$ and $b$ such that learning rate scheme of $\eta_t = b \cdot \exp\left(-at\right)$ achieves $\mathbb{E}\left[f(\mathbf{w}_T)\right] - f(\mathbf{w}^*) \leq \frac{f(\mathbf{w}_0) - f(\mathbf{w}^*)}{T^{2 - (1/100)}} + \log \kappa \cdot \log T \cdot \frac{\sigma^2 d}{T}$.*

The above results show that constant and cut as well as exponential decay schemes, that depend on the time horizon, are much better than polynomial decay schemes. Between these, exponential decay schemes are smoother versions of constant and cut schemes, and so one would hope that they might have better performance than constant and cut schemes – we do see a $\log T$ difference in our bounds. One unsatisfying aspect of the above results is that the rate behaves as $\frac{\log T}{T}$, which is asymptotically worse than the statistical rate of $\frac{1}{T}$. It turns out that it is indeed possible to improve the rate to $\frac{1}{T}$ using a more sophisticated scheme. The main idea is to use constant and polynomial schemes in the beginning and then switch to constant and cut (or exponential decay) scheme later. To the best of our knowledge, these kind of schemes have never been considered in the stochastic optimization literature before. Using this learning rate sequence successively for increasing time horizons would lead to oscillating learning rates. We leave a complete analysis of oscillating learning rates (for moving time horizon) to future work.

**Theorem 4.** *Fix $\kappa \geq 2$. For any problem and fixed time horizon $T/\log T > 5\kappa$, there exists a learning rate scheme that achieves $\mathbb{E}\left[f(\mathbf{w}_T)\right] - f(\mathbf{w}^*) \leq \frac{f(\mathbf{w}_0) - f(\mathbf{w}^*)}{T^3} + 50 \log_2 \kappa \cdot \frac{\sigma^2 d}{T}$.*

## 5 INFINITE HORIZON SETTING

In this section we show a fundamental limitation of the SGD algorithm. First we will prove that the SGD algorithm, for any learning rate sequence, needs to query a point with suboptimality more than $\Omega(\kappa/\log \kappa) \cdot \sigma^2 d/T$ for infinitely many time steps $T$.

**Theorem 5.** *There exists a universal constant $C > 0$ such that for any SGD algorithm with $\eta_t \leq 1/2\kappa$ for all $t$[9], we have $\limsup_{T \to \infty} \frac{\mathbb{E}[f(\mathbf{w}_T)] - f(\mathbf{w}^*)}{(\sigma^2 d/T)} \geq \frac{\kappa}{C \log(\kappa+1)}$.*

Next we will show that in some sense the "fraction" of query points that has value more than $\tau \sigma^2/T$ is at least $\Omega(1/\tau)$ when $\tau$ is smaller than the threshold in Theorem 5.

**Theorem 6.** *There exists universal constants $C_1, C_2 > 0$ such that for any $\tau \leq \frac{\kappa}{CC_1 \log(\kappa+1)}$ where $C$ is the constant in Theorem 5, for any SGD algorithm and any number of iteration $T > 0$ there exists a $T' \geq T$ such that for any $\tilde{T} \in [T', (1 + 1/C_2\tau)T']$ we have $\frac{\mathbb{E}\left[f(\mathbf{w}_{\tilde{T}})\right] - f(\mathbf{w}^*)}{\left(\sigma^2 d/\tilde{T}\right)} \geq \tau$.*

Finally, we now show that there are constant and cut or exponentially decaying schemes that achieve the statistical minimax rate up to a factor of $\log \kappa \log^2 T$ in the $\liminf$ sense.

---

[9]Learning rate more than $2/\kappa$ will make the algorithm diverge.

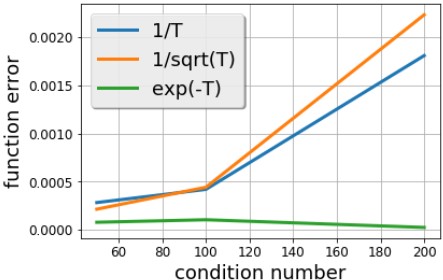

Figure 1: Plot of final error versus condition number $\kappa$ for decay schemes 1,2,3 for the 2-d regression problem.

| Model | err @$50\kappa$ | err @$100\kappa$ | err @$200\kappa$ |
|---|---|---|---|
| optimized for $50\kappa$ | **0.00018** | **$9.6 \times 10^{-5}$** | $9.6 \times 10^{-5}$ |
| optimized for $100\kappa$ | **0.00018** | **$9.6 \times 10^{-5}$** | $9.6 \times 10^{-5}$ |
| optimized for $200\kappa$ | 0.00275 | 0.0007 | **$2.5 \times 10^{-5}$** |

Table 1: Does the learning rate sequence optimized for a given end time generalize to other end times? Observe how a model optimized to perform for a specific horizon behaves sub-optimally for other time horizons.

**Theorem 7.** *There exists an absolute constant $C$ and a constant and cut learning rate scheme that obtains* $\liminf_{T \to \infty} \frac{\mathbb{E}[f(\mathbf{w}_T)] - f(\mathbf{w}^*)}{(\sigma^2 d \log^2 T/T)} \leq C \log \kappa$.

Similar results can be obtained for the exponential decay scheme of Theorems 3 and 4 with moving time horizon. However the resultant learning rates might have oscillatory behavior. This might partly explain the benefits of oscillating learning rates observed in practice (Smith & Topin, 2017).

## 6 EXPERIMENTAL RESULTS

We present experimental validation of our claims through controlled synthetic experiments on a two-dimensional quadratic objective and on a real world non-convex optimization problem of training a residual network on the cifar-10 dataset, to illustrate the shortcomings of the traditional stochastic approximation perspective (and the advantages of non-convergent exponentially decaying and oscillating learning rate schemes) for a realistic problem encountered in practice. Complete details of experimental setup are given in Appendix D.

### 6.1 SYNTHETIC EXPERIMENTS: TWO-DIMENSIONAL QUADRATIC OBJECTIVE

We consider the problem of optimizing a two-dimensional quadratic objective, similar in spirit as what is considered in the theoretical results of this paper. In particular, for a two-dimensional quadratic, we have two eigenvalues, one of magnitude $\kappa$ and the other being 1. We vary our condition number $\kappa \in \{50, 100, 200\}$ and use a total of $200\kappa$ iterations for optimization. The results expressed in this section are obtained by averaging over two random seeds. The learning rate schemes we search over are:

$$\eta_t = \frac{\eta_0}{1 + b \cdot t} \quad (1) \qquad \eta_t = \frac{\eta_0}{1 + b\sqrt{t}} \quad (2) \qquad \eta_t = \eta_0 \cdot \exp\left(-b \cdot t\right). \quad (3)$$

For the schemes detailed above, there are two parameters that need to be searched over: (i) the starting learning rate $\eta_0$ and, (ii) the decay factor $b$. We perform a grid search over both these parameters and choose ones that yield the best possible final error at a given end time (i.e. $200\kappa$). We also make sure to extend the grid should a best performing grid search parameter fall at the edge of the grid so that all presented results lie in the interior of our final grid searched parameters.

We will present results for the following experiments: (i) behavior of the error of the final iterate of the SGD method with the three learning rate schemes (1),(2), and (3) as we vary the condition number, and (ii) how the exponentially decaying learning rate scheme (3) optimized for a shorter time horizon behaves for a longer horizon.

For the variation of the final iterate's excess risk when considered with respect to the condition number (Figure 1), we note that polynomially decaying schemes have excess risk that scales linearly with condition number, corroborating Theorem 1. In contrast, exponentially decaying learning rate scheme admits excess risk that nearly appears to be a constant and corroborates Theorem 3. Finally, we note that the learning rate schedule that offers the best possible error in $50\kappa$ or $100\kappa$ steps does not offer the best error at $200\kappa$ steps (Table 1).

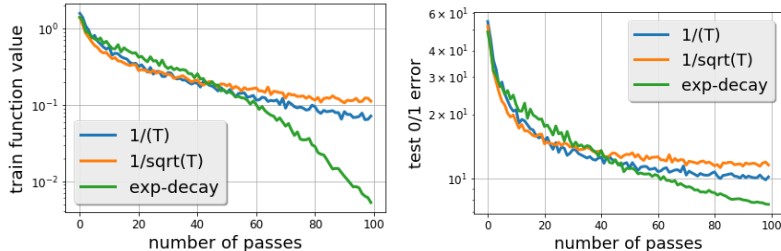

Figure 2: Plot of the training function value (left) and test $0/1-$ error (right) comparing the three decay schemes (two polynomial) 1, 2, (and one exponential) 3 scheme on the classification problem of cifar-10 with a $44-$layer residual net with pre-activation blocks.

## 6.2 NON-CONVEX OPTIMIZATION: TRAINING A RESIDUAL NET ON CIFAR-10

We consider here the task of training a $44-$layer deep residual network (He et al., 2016b) with pre-activation blocks (He et al., 2016a) (dubbed preresnet-44) for classifying images in the cifar-10 dataset. The code for implementing the network employed in this paper can be found here[10]. For all the experiments, we use the Nesterov's Accelerated gradient method (Nesterov, 1983) implemented in pytorch [11] with a momentum set to $0.9$ and batchsize set to $128$, total number of training epochs set to $100$, $\ell_2$ regularization set to $0.0005$.

Our experiments are based on grid searching for the best learning rate decay scheme on four parametric family of learning rate schemes described above 1,2,3; all gridsearches are performed on a separate validation set (obtained by setting aside one-tenth of the training dataset $= 5000$ images) and with models trained on the remaining $45000$ images. For presenting the final numbers in the plots/tables, we employ the best hyperparameters from the validation stage and train it on the entire $50,000$ images and average results run with $10$ different random seeds. The parameters for gridsearches and related details are presented in Appendix D. Furthermore, just as with the synthetic experiments, we always extend the grid so that the best performing grid search parameter lies in the interior of our grid search.

**Comparison between different schemes**: Figure 2 and Table 2 present a comparison of the performance of the three schemes (1)-(3). They clearly demonstrate that the best exponential scheme outperforms the best polynomial schemes.

**Hyperparameter selection using truncated runs**: Figure 3 and Tables 3 and 4 present a comparison of the performance of three exponential decay schemes each of which has the best performance at 33, 66 and 100 epochs respectively. The key point to note is that best performing hyperparameters at 33 and 66 epochs are not the best performing at 100 epochs (which is made stark from the perspective of the validation error). This demonstrates that selecting hyper parameters using truncated runs, which has been proposed in some recent efforts such as hyperband (Li et al., 2017), might necessitate rethinking.

| Decay Scheme | Train Function Value | Test $0/1$ error |
|---|---|---|
| $O(1/t)$ (equation 1) | $0.0713 \pm 0.015$ | $10.20 \pm 0.7\%$ |
| $O(1/\sqrt{t})$ (equation 2) | $0.1119 \pm 0.036$ | $11.6 \pm 0.67\%$ |
| $\exp(-t)$ (equation 3) | $\mathbf{0.0053 \pm 0.0015}$ | $\mathbf{7.58 \pm 0.21\%}$ |

Table 2: Comparing Train Softmax Function Value and Test $0/1$ Error of various learning rate decay schemes for the classification task on cifar-10 using a $44-$layer residual net with pre-activations.

---

[10]https://github.com/D-X-Y/ResNeXt-DenseNet
[11]https://github.com/pytorch

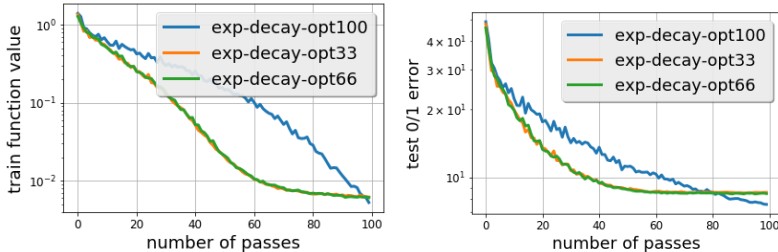

Figure 3: Plot of the training function value (left) and test $0/1-$ error (right) comparing the exponential decay scheme (equation 3), with parameters optimized for 33, 66 and 100 epochs on the classification problem of cifar-10 with a $44-$layer residual net with pre-activation blocks.

| Decay Scheme | Train FVal @33 | Train FVal @66 | Train FVal @100 |
|---|---|---|---|
| $\exp(-t)$ [optimized for 33 epochs] (eqn 3) | **0.098 ± 0.006** | **0.0086 ± 0.002** | 0.0062 ± 0.0015 |
| $\exp(-t)$ [optimized for 66 epochs] (eqn 3) | **0.107 ± 0.012** | **0.0088 ± 0.0014** | 0.0061 ± 0.0011 |
| $\exp(-t)$ [optimized for 100 epochs] (eqn 3) | 0.3 ± 0.06 | 0.071 ± 0.017 | 0.0053 ± 0.0016 |

Table 3: Comparing Train Softmax Function Value of various learning rate decay schemes for the classification task on cifar-10 using a $44-$layer residual net with pre-activations.

| Decay Scheme | Test 0/1 @33 | Test 0/1 @66 | Test 0/1 @100 |
|---|---|---|---|
| $\exp(-t)$ [optimized for 33 epochs] (eqn 3) | **10.36 ± 0.235%** | **8.6 ± 0.26%** | 8.57 ± 0.25% |
| $\exp(-t)$ [optimized for 66 epochs] (eqn 3) | **10.51 ± 0.45%** | **8.51 ± 0.13%** | 8.46 ± 0.19% |
| $\exp(-t)$ [optimized for 100 epochs] (eqn 3) | 14.42 ± 1.47% | 9.8 ± 0.66% | **7.58 ± 0.21%** |

Table 4: Comparing Test $0/1$ error of various learning rate decay schemes for the classification task on cifar-10 using a $44-$layer residual net with pre-activations.

## 7 CONCLUSIONS AND DISCUSSION

The main contribution of this work shows that the picture of learning rate scheduling is far more nuanced than suggested by prior theoretical results, where we do not even need to move to non-convex optimization to show other learning rate schemes can be far more effective than the standard polynomially decaying rates considered in theory.

**Is quadratic loss minimization special?** One may ask if there is something particularly special about why the minimax rates are different for quadratic loss minimization as opposed to more general convex (and non-convex) optimization problems? Ideally, we would hope that our theoretical insights (and improvements) can be formally established in more general cases. Here, an alternative viewpoint is to consider gradient norm as a means to measure the progress of an algorithm. The recent work of Allen-Zhu (2018) shows marked improvements for making the gradient norm small (when working with stochastic gradients) for both convex and non-convex, in comparison to prior results. In particular, for the strongly convex case, Allen-Zhu (2018) provides results which have only a logarithmic dependency on $\kappa$, an exponential improvement over what is implied by standard analyses for the gradient norm (Lacoste-Julien et al., 2012; Rakhlin et al., 2012; Bubeck, 2014); Allen-Zhu (2018) also provides improvements for the smooth and non-convex cases. Thus, for the case of making the gradient norm small, there does not appear to be a notable discrepancy between the minimax rate of quadratic loss minimization in comparison to more general strongly convex (or smooth) convex optimization problems. Interestingly, the algorithm of Allen-Zhu (2018) provides a recursive regularization procedure that obtains an SGD procedure, where the doubling regularization can be viewed as being analogous to an exponentially decaying learning rate schedule. Further work in this direction may be promising in providing improved algorithms.

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

## A  PROOFS OF RESULTS IN SECTION 4.1

*Proof of Theorem 1.* The problem instance is simple. Let $\mathbf{H} = \begin{bmatrix} \kappa & & & & \\ & \ddots & & & \\ & & 1 & & \\ & & & \ddots & \end{bmatrix}$, where the first

$\frac{d}{2}$ diagonal entries are equal to $\kappa$ and the remaining $\frac{d}{2}$ diagonal entries are equal to 1 and all the off diagonal entries are equal to zero. Let us denote by $v_t^{(i)} \stackrel{\text{def}}{=} \mathbb{E}\left[\left(\mathbf{w}_t^{(i)} - (\mathbf{w}^*)^{(i)}\right)^2\right]$ the variance in the $i^{\text{th}}$ direction at time step $t$. Let the initialization be such that $v_0^{(i)} = \sigma^2/\kappa$ for $i = 1, 2, ..., d/2$ and $v_0^{(i)} = \sigma^2$ for $i = d/2+1, ..., d$. This means that the variances for all directions with eigenvalue $\kappa$ remain equal as $t$ progresses and similarly for all directions with eigenvalue 1. We have

$$v_T^{(1)} \stackrel{\text{def}}{=} \mathbb{E}\left[\left(\mathbf{w}_T^{(1)} - (\mathbf{w}^*)^{(1)}\right)^2\right] = \prod_{j=1}^{T}(1 - \eta_j\kappa)^2 v_0^{(1)} + \kappa\sigma^2\sum_{j=1}^{T}\eta_j^2\prod_{i=j+1}^{T}(1 - \eta_i\kappa)^2 \text{ and}$$

$$v_T^{(d)} \stackrel{\text{def}}{=} \mathbb{E}\left[\left(\mathbf{w}_T^{(d)} - (\mathbf{w}^*)^{(d)}\right)^2\right] = \prod_{j=1}^{T}(1 - \eta_j)^2 v_0^{(d)} + \sigma^2\sum_{j=1}^{T}\eta_j^2\prod_{i=j+1}^{T}(1 - \eta_i)^2.$$

We consider a recursion for $v_t^{(i)}$ with eigenvalue $\lambda_i$ (1 or $\kappa$). By the design of the algorithm, we know

$$v_{t+1}^{(i)} = (1 - \eta_t\lambda_i)^2 v_t^{(i)} + \lambda_i\sigma^2\eta_t^2.$$

Let $s(\eta, \lambda) = \frac{\lambda\sigma^2\eta^2}{1-(1-\eta\lambda)^2}$ be the solution to the stationary point equation $x = (1 - \eta\lambda)^2 + \lambda\sigma^2\eta^2$. Intuitively if we keep using the same learning rate $\eta$, then $v_t^{(i)}$ is going to converge to $s(\eta, \lambda_i)$. Also note that $s(\eta, \lambda) \approx \sigma^2\eta/2$ when $\eta\lambda \ll 1$.

We first prove the following claim showing that eventually the variance in direction $i$ is going to be at least $s(\eta_T, \lambda_i)$.

**Claim 1.** *Suppose $s(\eta_t, \lambda_i) \leq v_0^{(i)}$, then $v_t^{(i)} \geq s(\eta_t, \lambda_i)$.*

*Proof.* We can rewrite the recursion as

$$v_{t+1}^{(i)} - s(\eta_t, \lambda_i) = (1 - \eta_t\lambda_i)^2(v_t^{(i)} - s(\eta_t, \lambda_i)).$$

In this form, it is easy to see that the iteration is a contraction towards $s(\eta_t, \lambda_i)$. Further, $v_{t+1}^{(i)} - s(\eta_t, \lambda_i)$ and $v_t^{(i)} - s(\eta_t, \lambda_i)$ have the same sign. In particular, let $t_0$ be the first time such that $s(\eta_t, \lambda_i) \leq v_0^{(i)}$ (note that $\eta_t$ is monotone and so is $s(\eta_t, \lambda_i)$), it is easy to see that $v_t^{(i)} \geq v_0^{(i)}$ when $t \leq t_0$. Therefore we know $v_{t_0}^{(i)} \geq s(\eta_{t_0}, \lambda_i)$, by the recursion this implies $v_{t_0+1}^{(i)} \geq s(\eta_{t_0}, \lambda_i) \geq s(\eta_{t_0+1}, \lambda_i)$. The claim then follows from a simple induction. □

If $s(\eta_T, \lambda_i) \geq v_0^{(i)}$ for $i = 1$ or $i = d$ then the error is at least $\sigma^2 d/2 \geq \kappa\sigma^2 d/T$ and we are done. Therefore we must have $s(\eta_T, \kappa) \leq v_0^{(1)} = \sigma^2/\kappa$, and by Claim 1 we know $v_T^{(1)} \geq s(\eta_T, \kappa) \geq \sigma^2\eta_T/2$. The function value is at least

$$\mathbb{E}\left[f(\mathbf{w}_T)\right] \geq \frac{d}{2} \cdot \kappa v_T^{(1)} \geq \frac{d\kappa\sigma^2\eta_T}{4}.$$

To make sure $\mathbb{E}\left[f(\mathbf{w}_T)\right] \leq \frac{d\kappa\sigma^2\eta_T}{32}$ we must have $\eta_T \leq \frac{1}{8T}$. Next we will show that when this happens, $v_T^{(d)}$ must be large so the function value is still large.

We will consider two cases, in the first case, $b \geq T^\alpha$. Since $\frac{1}{8T} \geq \eta_T = \frac{a}{b+T^\alpha} \geq \frac{a}{2b}$, we have $\frac{a}{b} \leq \frac{1}{4T}$. Therefore $v_T^{(d)} \geq (1 - \frac{a}{b})^{2T} v_0^{(d)} \geq \sigma^2/2$, so the function value is at least $\mathbb{E}\left[f(\mathbf{w}_t)\right] \geq \frac{d}{2}v_T^{(d)} \geq \frac{d\sigma^2}{4} \geq \frac{\kappa d\sigma^2}{T}$, and we are done.

In the second case, $b < T^\alpha$. Since $\frac{1}{8T} \ge \eta_T = \frac{a}{b+T^\alpha} \ge \frac{a}{2T^\alpha}$, we have $a \le 0.25T^{\alpha-1}$. The sum of learning rates satisfy

$$\sum_{i=1}^{T} \eta_i \le \sum_{i=1}^{T} \frac{a}{i^\alpha} \le \sum_{i=1}^{T} 0.25i^{-1} \approx 0.25 \log T.$$

Here the second inequality uses the fact that $T^{\alpha-1}i^{-\alpha} \le i^{-1}$ when $i \le T$. Similarly, we also know $\sum_{i=1}^{T} \eta_i^2 \le \sum_{i=1}^{T} 0.25i^{-2} \le \pi^2/24$. Using the approximation $(1-\eta)^2 \ge \exp(-2\eta - 4\eta^2)$ for $\eta < 1/4$, we get $v_T^{(d)} \ge \exp(-2\sum_{i=1}^{T} \eta_i - 4\sum_{i=1}^{T} \eta_i^2)v_0^{(d)} \ge \sigma^2/5\sqrt{T}$, so the function value is at least $\mathbb{E}\left[f(\mathbf{w}_t)\right] \ge \frac{d}{2}v_T^{(d)} \ge \frac{d\sigma^2}{20\sqrt{T}} \ge \frac{\kappa d\sigma^2}{32T}$. This concludes the second case and proves the theorem. $\qquad\square$

## B    PROOFS OF RESULTS IN SECTION 4.2

*Proof of Theorem 2.* The learning rate scheme is as follows. Divide the total time horizon into $\log(\kappa)$ equal sized phases. In the $\ell^{\text{th}}$ phase, the learning rate to be used is $\frac{\log T \cdot \log \kappa}{2^\ell \cdot \mu \cdot T}$. Note that the learning rate in the first phase depends on strong convexity and that in the last phase depends on smoothness (since the last phase has $\ell = \log \kappa$). Recall the variance in the $k^{\text{th}}$ coordinate can be upper bounded by

$$v_T^{(k)} \overset{\text{def}}{=} \mathbb{E}\left[\left(\mathbf{w}_T^{(k)} - (\mathbf{w}^*)^{(1)}\right)^2\right] \le \prod_{j=1}^{T}\left(1 - \eta_j\lambda^{(k)}\right)^2 v_0^{(1)} + \lambda^{(k)}\sigma^2 \sum_{j=1}^{T}\eta_j^2 \prod_{i=j+1}^{T}\left(1 - \eta_i\lambda^{(k)}\right)^2$$

$$\le \exp\left(-2\sum_{j=1}^{T}\eta_j\lambda^{(k)}\right)v_0^{(1)} + \lambda^{(k)}\sigma^2 \sum_{j=1}^{T}\eta_j^2 \exp\left(-2\sum_{i=j+1}^{T}\eta_i\lambda^{(k)}\right).$$

We will show that for every $k$, we have

$$v_T^{(k)} \le \frac{\log\kappa \cdot \log T}{\lambda^{(k)}T}\cdot\sigma^2,$$

which directly implies the theorem. Now choose any $k$. Let $\ell^*$ denote the number satisfying $2^{\ell^*}\cdot\mu \le \lambda^{(k)} < 2^{\ell^*+1}\cdot\mu$. Note that $\ell^*$ depends on $k$ but we suppressed the dependence for notational simplicity.

$$v_T^{(k)} \le \exp\left(-2\sum_{j=1}^{T}\eta_j\lambda^{(k)}\right)v_0^{(k)} + \lambda^{(k)}\sigma^2 \sum_{j=1}^{T}\eta_j^2 \exp\left(-2\sum_{i=j+1}^{T}\eta_i\lambda^{(k)}\right)$$

$$\le \exp\left(-2\frac{\log T \log \kappa}{T}\cdot\lambda^{(k)}\cdot\frac{T}{\log\kappa}\cdot\frac{1}{2^{\ell^*}\mu}\right)\left(v_0^{(k)} + \lambda^{(k)}\sigma^2 \cdot \frac{T}{\log\kappa}\cdot\sum_{\ell=1}^{\ell^*-1}\eta_\ell^2\right)$$

$$+ \lambda^{(k)}\sigma^2\left(\eta_{\ell^*}^2 \cdot \frac{1}{1-\exp\left(-\eta_{\ell^*}\lambda^{(k)}\right)} + \sum_{\ell=\ell^*+1}^{\log\kappa}\frac{T}{\log\kappa}\cdot\left(\frac{\log\kappa\log T}{2^\ell\mu T}\right)^2\right)$$

$$\le \frac{v_0^{(k)}}{T^3} + \kappa\sigma^2 \cdot \frac{\log^2\kappa\log^2 T}{\mu T^3} + \eta_{\ell^*}\sigma^2 + \sigma^2 \cdot \frac{\log\kappa\log^2 T}{T}\sum_{\ell=\ell^*+1}^{\log\kappa}\frac{1}{2^\ell\mu}$$

$$\le \frac{v_0^{(k)}}{T^3} + \sigma^2 \cdot \frac{\log\kappa\log^2 T}{T}\left(\lambda^{(k)} + \frac{1}{2^{\ell^*}\mu}\right)$$

$$\le \frac{v_0^{(k)}}{T^3} + \frac{2\log\kappa\cdot\log^2 T}{\lambda^{(k)}T}\cdot\sigma^2.$$

This finishes the proof. $\qquad\square$

*Proof of Theorem 3.* The learning rate scheme we consider is $\gamma_t = \gamma_0 \cdot c^{t-1}$ with $\gamma_0 = \log T/(\mu T_e)$, $T_e = T/\log \kappa$ and $c = (1 - 1/T_e)$. Further, just as in previous lemmas, we consider a specific eigen direction $\lambda^{(k)}$ and write out the progress made along this direction by iteration say, $\hat{T} \leq T$:

$$\text{err}_{\hat{T}}^{(k)} = \prod_{t=1}^{\hat{T}}(1 - \gamma_t \lambda^{(k)})^2 \text{err}_0^{(k)} + (\lambda^{(k)})^2 \sigma^2 \sum_{\tau=1}^{\hat{T}} \gamma_\tau^2 \cdot \left( \prod_{t=\tau+1}^{\hat{T}} (1 - \gamma_t \lambda^{(k)})^2 \right)$$

$$\leq \exp\left( -2\lambda^{(k)} \gamma_0 \sum_{t=1}^{\hat{T}} c^{t-1} \right) \text{err}_0^{(k)} + \frac{(\lambda^{(k)})^2 \sigma^2 \gamma_0^2}{c^2} \sum_{\tau=1}^{\hat{T}} c^{2\tau} \exp\left( -2\lambda^{(k)} \gamma_0 \sum_{t=\tau+1}^{\hat{T}} c^{t-1} \right)$$

$$= \exp\left( -\frac{2\lambda^{(k)} \gamma_0}{1-c} \cdot (1 - c^{\hat{T}}) \right) \text{err}_0^{(k)} + \frac{(\lambda^{(k)})^2 \sigma^2 \gamma_0^2}{c^2} \sum_{\tau=1}^{\hat{T}} c^{2\tau} \exp\left( -\frac{2\lambda^{(k)} \gamma_0}{1-c} \cdot (c^\tau - c^{\hat{T}}) \right)$$

$$= \exp\left( \frac{2\lambda^{(k)} \gamma_0}{1-c} \cdot c^{\hat{T}} \right) \cdot \left( \exp\left( -\frac{2\lambda^{(k)} \gamma_0}{1-c} \right) \text{err}_0^{(k)} + \frac{(\lambda^{(k)})^2 \sigma^2 \gamma_0^2}{c^2} \sum_{\tau=1}^{\hat{T}} c^{2\tau} \exp\left( -\frac{2\lambda^{(k)} \gamma_0}{1-c} \cdot (c^\tau) \right) \right)$$

Represent $c^\tau = x$ and $2\lambda^{(k)} \gamma_0 / (1-c) = \alpha$. Substituting necessary values of the quantities, we have $\alpha = 2\log T \cdot \lambda^{(k)}/\mu \geq 2$. Now, the second term is upper bounded using the corresponding integral, which is the following:

$$\sum_{x=c}^{c^{\hat{T}}} x^2 \exp(-\alpha x) \leq \int_1^{c^{\hat{T}}} x^2 \exp(-\alpha x)dx \leq \frac{1}{\alpha} \cdot \left( 1 + \frac{2}{\alpha} + \frac{2}{\alpha^2} \right) \exp(-\alpha).$$

Substituting this in the previous bound, we have:

$$\text{err}_{\hat{T}}^{(k)} \leq \exp\left( -\alpha(1 - c^{\hat{T}}) \right) \cdot \left( \text{err}_0^{(k)} + \frac{(\lambda^{(k)})^2 \sigma^2 \gamma_0^2}{\alpha c^2} \cdot \left( 1 + \frac{\sqrt{2}}{\alpha} \right)^2 \right)$$

$$\leq \exp\left( -\alpha(1 - c^{\hat{T}}) \right) \cdot \left( \text{err}_0^{(k)} + 16\frac{(\lambda^{(k)})^2 \sigma^2 \gamma_0^2}{\alpha} \right)$$

$$\leq \exp\left( -\alpha(1 - c^{\hat{T}}) \right) \cdot \left( \text{err}_0^{(k)} + 16\frac{(\lambda^{(k)})^2 \sigma^2 \log T}{\mu^2 T_e^2 \cdot 2(\lambda^{(k)}/\mu)} \right)$$

$$= \exp\left( -\alpha(1 - c^{\hat{T}}) \right) \cdot \left( \text{err}_0^{(k)} + 8\frac{\lambda^{(k)} \sigma^2 \log T}{\mu T_e^2} \right)$$

Now, setting $\hat{T} = T_e \log(C\lambda^{(k)}/\mu)$, and using $1 - a \leq \exp(-a)$, with $C > 1$ being some (large) universal constant, we have:

$$\text{err}_{\hat{T}}^{(k)} \leq \exp\left( -2(\lambda^{(k)}/\mu) \log T \cdot \left( 1 - \frac{\mu}{C\lambda^{(k)}} \right) \right) \cdot \left( \text{err}_0^{(k)} + 8\frac{\lambda^{(k)} \sigma^2 \log T}{\mu T_e^2} \right)$$

$$\leq \frac{1}{T^{2-(1/C)}} \cdot \left( \text{err}_0^{(k)} + 8\frac{\lambda^{(k)} \sigma^2 \log T}{\mu T_e^2} \right) \tag{4}$$

Now, in order to argue the progress of the algorithm from $\hat{T} + 1$ to $T$, we can literally view the algorithm as starting with the iterate obtained from running for the first $\hat{T}$ steps (thus satisfying the excess risk guarantee in equation 4) and then adding in variance by running for the duration of time between $\hat{T} + 1$ to $T$. For this part, we basically upper bound this behavior by first assuming that there is no contraction of the bias and then consider the variance introduced by running the algorithm from $\hat{T} + 1$ to $T$. This can be written as:

$$\text{err}_T^{(k)} \leq \prod_{t=\hat{T}+1}^{T} (1 - \gamma_t \lambda^{(k)})^2 \text{err}_{\hat{T}}^{(k)} + (\lambda^{(k)})^2 \sigma^2 \sum_{\tau=1}^{T-\hat{T}} \gamma_{\tau+\hat{T}}^2 \prod_{t=\tau+1}^{T-\hat{T}} (1 - \gamma_{t+\hat{T}} \lambda^{(k)})^2$$

$$\leq \text{err}_{\hat{T}}^{(k)} + (\lambda^{(k)})^2 \sigma^2 \sum_{\tau=1}^{T-\hat{T}} \gamma_{\tau+\hat{T}}^2 \prod_{t=\tau+1}^{T-\hat{T}} (1 - \gamma_{t+\hat{T}} \lambda^{(k)})^2$$

Now, to consider bounding the variance of the process with decreasing sequence of learning rates, we will instead work with a constant learning rate and understand its variance:

$$(\lambda^{(k)})^2 \sigma^2 \sum_{\tau=1}^{T-\hat{T}} \gamma^2 (1 - \gamma \lambda^{(k)})^{2(T-\hat{T}-\tau)} \leq \frac{\gamma^2 \sigma^2 (\lambda^{(k)})^2}{(2 - \lambda^{(k)} \gamma) \gamma \lambda^{(k)}}$$

$$\leq \gamma \sigma^2 \lambda^{(k)}.$$

What this implies in particular is that the variance is a monotonic function of the learning rate and thus the overall variance can be bounded using the variance of the process run with a learning rate of $\gamma_{\hat{T}}$.

$$(\lambda^{(k)})^2 \sigma^2 \sum_{\tau=1}^{T-\hat{T}} \gamma_{\tau+\hat{T}}^2 \prod_{t=\tau+1}^{T-\hat{T}} (1 - \gamma_{t+\hat{T}} \lambda^{(k)})^2 \leq (\lambda^{(k)})^2 \sigma^2 \sum_{t=\tau+1}^{T-\hat{T}} \gamma_{\hat{T}}^2 (1 - \gamma_{\hat{T}} \lambda^{(k)})^2 \leq \gamma_{\hat{T}} \sigma^2 \lambda^{(k)}$$

$$\leq \frac{\log T}{\mu T_e} \cdot \frac{\mu}{C \lambda^{(k)}} \cdot \sigma^2 \lambda^{(k)}$$

$$\leq \frac{\sigma^2 \log T \log \kappa}{T}$$

Plugging this into equation 4 and summing over all directions, we have the desired result. $\qquad\square$

*Proof of Theorem 4.* The learning rate scheme is as follows.

We first break $T$ into three equal sized parts. Let $A = T/3$ and $B = 2T/3$. In the first $T/3$ steps, we use a constant learning rate of $1/L$. In the second $T/3$ steps, we use a polynomial decay learning rate $\eta_{A+t} = \frac{1}{\mu(\kappa+t/2)}$. In the third $T/3$ steps, we break the steps into $\log_2(\kappa)$ equal sized phases. In the $\ell^{\text{th}}$ phase, the learning rate to be used is $\frac{5 \log_2 \kappa}{2^\ell \cdot \mu \cdot T}$. Note that the learning rate in the first phase depends on strong convexity and that in the last phase depends on smoothness (since the last phase has $\ell = \log \kappa$).

Recall the variance in the $k^{\text{th}}$ coordinate can be upper bounded by

$$v_T^{(k)} \stackrel{\text{def}}{=} \mathbb{E}\left[\left(\mathbf{w}_T^{(k)} - (\mathbf{w}^*)^{(1)}\right)^2\right] \leq \prod_{j=1}^T \left(1 - \eta_j \lambda^{(k)}\right)^2 v_0^{(1)} + \lambda^{(k)} \sigma^2 \sum_{j=1}^T \eta_j^2 \prod_{i=j+1}^T \left(1 - \eta_i \lambda^{(k)}\right)^2$$

$$\leq \exp\left(-2 \sum_{j=1}^T \eta_j \lambda^{(k)}\right) v_0^{(1)} + \lambda^{(k)} \sigma^2 \sum_{j=1}^T \eta_j^2 \exp\left(-2 \sum_{i=j+1}^T \eta_i \lambda^{(k)}\right).$$

We will show that for every $k$, we have

$$v_T^{(k)} \leq \frac{v_0^{(k)}}{T^3} + \frac{50 \log_2 \kappa}{\lambda^{(k)} T} \cdot \sigma^2., \tag{5}$$

which directly implies the theorem.

We will consider the first $T/3$ steps. The guarantee that we will prove for these iterations is: for any $t \leq A$, $v_t^{(k)} \leq (1 - \lambda^{(k)}/L)^{2t} v_0^{(k)} + \frac{\sigma^2}{L}$.

This can be proved easily by induction. Clearly this is true when $t = 0$. Suppose it is true for $t-1$, let's consider step $t$. By recursion of $v_t^{(k)}$ we know

$$v_t^{(k)} = (1 - \lambda^{(k)}/L)^2 v_{t-1}^{(k)} + \lambda^{(k)} \sigma^2 / L^2$$

$$\leq (1 - \lambda^{(k)}/L)^{2t} v_0^{(k)} + \frac{\sigma^2}{L}\left((1 - \lambda^{(k)}/L)^2 + \lambda^{(k)}/L\right)$$

$$\leq (1 - \lambda^{(k)}/L)^{2t} v_0^{(k)} + \frac{\sigma^2}{L}.$$

Here the second step uses induction hypothesis and the third step uses the fact that $(1-x)^2 + x \leq 1$ when $x \in [0, 1]$. In particular, since $(1 - \lambda^{(k)}/L)^{2T/3} \leq (1 - 1/\kappa)^{2T/3} \leq (1 - 1/\kappa)^{3\kappa \log T} = 1/T^3$, we know at the end of the first phase, $v_A^{(k)} \leq v_0^{(k)}/T^3 + \frac{\sigma^2}{L}$.

In the second $T/3$ steps, the guarantee would be: for any $t \leq T/3$, $v_{A+t}^{(k)} \leq v_0^{(k)}/T^3 + 2\eta_{A+t}\sigma^2$.

We will again prove this by induction. The base case ($t = 0$) follows immediately from the guarantee for the first part. Suppose this is true for $A + t - 1$, let us consider $A + t$, again by recursion we know

$$
\begin{aligned}
v_{A+t}^{(k)} &= (1 - \lambda^{(k)}\eta_{A+t-1})^2 v_{A+t-1}^{(k)} + \lambda^{(k)}\sigma^2\eta_{A+t-1}^2 \\
&\leq v_0^{(k)}/T^3 + 2\eta_{A+t-1}\sigma^2\left((1 - \lambda^{(k)}\eta_{A+t-1})^2 + \frac{1}{2}\lambda^{(k)}\eta_{A+t-1}\right) \\
&\leq v_0^{(k)}/T^3 + 2\eta_{A+t-1}\sigma^2(1 - \frac{1}{2}\mu\eta_{A+t-1}) \leq v_0^{(k)}/T^3 + 2\eta_{A+t}\sigma^2.
\end{aligned}
$$

Here the last line uses the fact that $2\eta_{A+t-1}(1 - \frac{1}{2}\mu\eta_{A+t-1}) \leq 2\eta_{A+t}\sigma^2$, which is easy to verify by our choice of $\eta$. Therefore, at the end of the second part, we have $v_B^{(k)} \leq v_0^{(k)}/T^3 + \frac{2\sigma^2}{\mu(\kappa+T/6)}$.

Finally we will analyze the third part. Let $\hat{T} = T/3\log_2\kappa$, we will consider the variance $v_{B+\ell\hat{T}}^{(k)}$ at the end of each phase. We will make the following claim by induction:

**Claim 2.** *Suppose $2^\ell \cdot \mu \leq \lambda^{(k)}$, then*

$$
v_{B+\ell\hat{T}}^{(k)} \leq v_B^{(k)}\exp(-3\ell) + 2\hat{T}\eta_\ell^2\lambda^{(k)}\sigma^2.
$$

*Proof.* We will prove this by induction. When $\ell = 0$, clearly we have $v_B^{(k)} \leq v_B^{(k)}$ so the claim is true. Suppose the claim is true for $\ell - 1$, we will consider what happens after the algorithm uses $\eta_\ell$ for $\hat{T}$ steps. By the recursion of the variance we have

$$
v_{\ell\hat{T}}^{(k)} \leq v_{(\ell-1)\hat{T}}^{(k)} \cdot \exp(-2\eta_\ell \cdot \lambda^{(k)}\hat{T}) + \hat{T}\eta_\ell^2\lambda^{(k)}\sigma^2.
$$

Since $2^\ell \cdot \mu \leq \lambda^{(k)}$, we know $\exp(-2\eta_\ell \cdot \lambda^{(k)}\hat{T}) \leq \exp(-3)$. Therefore by induction hypothesis we have

$$
v_{B+\ell\hat{T}}^{(k)} \leq v_B^{(k)}\exp(-3\ell) + \exp(-3) \cdot 2\hat{T}\eta_{\ell-1}^2\lambda^{(k)} + \hat{T}\eta_\ell^2\lambda^{(k)} \leq v_B^{(k)}\exp(-3\ell) + 2\hat{T}\eta_\ell^2\lambda^{(k)}.
$$

This finishes the induction. $\qquad\square$

By Claim 2, Let $\ell^*$ denote the number satisfying $2^{\ell^*} \cdot \mu \leq \lambda^{(k)} < 2^{\ell^*+1} \cdot \mu$, by this choice we know $\mu/\lambda^{(k)} \geq \frac{1}{2}\exp(-3\ell^\star)$ we have

$$
\begin{aligned}
v_T^{(k)} \leq v_{B+\ell^*\hat{T}}^{(k)} &\leq v_B^{(k)}\exp(-3\ell^*) + 2\hat{T}\eta_{\ell^*}^2\lambda^{(k)}\sigma^2 \\
&\leq \frac{v_0^{(k)}}{T^3} + \frac{24\sigma^2}{\lambda^{(k)}T} + \frac{50\log_2\kappa}{3\lambda^{(k)}T} \cdot \sigma^2. \\
&\leq \frac{v_0^{(k)}}{T^3} + \frac{50\log_2\kappa}{\lambda^{(k)}T} \cdot \sigma^2.
\end{aligned}
$$

Therefore, the function value is bounded by $\mathbb{E}\left[f(\mathbf{w}_T)\right] = \sum_{i=1}^d \lambda^{(k)}v_T^{(k)} \leq \frac{f(\mathbf{w}_0)}{T^3} + \frac{50\log_2\kappa}{T} \cdot \sigma^2 d$. $\qquad\square$

## C  PROOFS OF RESULTS IN SECTION 5

All of our counter-examples in this section are going to be the same simple function. Let $\mathbf{H}$ be a diagonal matrix with $d/2$ eigenvalues equal to $\kappa$ and the other $d/2$ eigenvalues equal to $1$. Intuitively, we will show that in order to have a small error in the first eigendirection (with eigenvalue $\kappa$), one need to set a small learning rate $\eta_t$ which would be too small to achieve a small error in the second

eigendirection (with eigenvalue 1). As a useful tool, we will decompose the variance in the two directions corresponding to $\kappa$ eigenvalue and 1 eigenvalue respectively as follows:

$$
v_T^{(1)} \stackrel{\text{def}}{=} \mathbb{E}\left[\left(\mathbf{w}_T^{(1)} - (\mathbf{w}^*)^{(1)}\right)^2\right] = \prod_{j=1}^{T}(1-\eta_j\kappa)^2 v_0^{(1)} + \kappa\sigma^2 \sum_{j=1}^{T}\eta_j^2 \prod_{i=j+1}^{T}(1-\eta_i\kappa)^2
$$

$$
\geq \exp\left(-2\sum_{j=1}^{T}\eta_j\kappa\right)v_0^{(1)} + \kappa\sigma^2\sum_{j=1}^{T}\eta_j^2\exp\left(-2\sum_{i=j+1}^{T}\eta_i\kappa\right) \text{ and} \tag{6}
$$

$$
v_T^{(2)} \stackrel{\text{def}}{=} \mathbb{E}\left[\left(\mathbf{w}_T^{(2)} - (\mathbf{w}^*)^{(2)}\right)^2\right] = \prod_{j=1}^{T}(1-\eta_j)^2 v_0^{(2)} + \sigma^2 \sum_{j=1}^{T}\eta_j^2 \prod_{i=j+1}^{T}(1-\eta_i)^2
$$

$$
\geq \exp\left(-2\sum_{j=1}^{T}\eta_j\right)v_0^{(2)} + \sigma^2\sum_{j=1}^{T}\eta_j^2\exp\left(-2\sum_{i=j+1}^{T}\eta_i\right). \tag{7}
$$

*Proof of Theorem 5.* Fix $\tau = \kappa/C\log(\kappa+1)$ where $C$ is a universal constant that we choose later. We need to exhibit that the $\limsup$ is larger than $\tau$. For simplicity we will also round $\kappa$ up to the nearest integer.

Let $T$ be a given number. Our goal is to exhibit a $\tilde{T} > T$ such that $\frac{f_{\text{lin}}(\mathbf{w}_{\tilde{T}})}{(\sigma^2/\tilde{T})} \geq \tau$. Given the step size sequence $\eta_t$, consider the sequence of numbers $T_0 = T, T_1, \cdots, T_\kappa$ such that $T_i$ is the first number that

$$
\frac{1}{\kappa} \leq \sum_{t=T_{i-1}+1}^{T_i}\eta_t \leq \frac{3}{\kappa}.
$$

Note that such a number always exists because all the step sizes are at most $2/\kappa$. We will also let $\Delta_i$ be $T_i - T_{i-1}$. Firstly, from (6) and (7), we see that $\sum_t \eta_t = \infty$. Otherwise, the bias will never decay to zero. If $f(\mathbf{w}_{T_{i-1}+\Delta_i}) > \frac{\tau\sigma^2 d}{T_{i-1}+\Delta_i}$ for some $i = 1, \cdots, \kappa$, we are done. If not, we obtain the following relations:

$$
\frac{\sigma^2}{\Delta_1} \leq \sigma^2\sum_{t=1}^{\Delta_1}\eta_{T_0+t}^2 \leq \frac{\exp(3)}{\kappa}\cdot\mathbb{E}\left[\left(\mathbf{w}_{T_0+\Delta_1}^{(1)} - (\mathbf{w}^*)^{(1)}\right)^2\right] \leq \exp(3)f_{\text{lin}}(\mathbf{w}_{T_0+\Delta_1}) \leq \frac{\exp(3)\tau\sigma^2}{T_0+\Delta_1}
$$

$$
\Rightarrow T_0 \leq (\exp(3)\tau - 1)\Delta_1.
$$

Here the second inequality is based on (6). We will use $C_1$ to denote $\exp(3)$. Similarly, we have

$$
\frac{\sigma^2}{\Delta_2} \leq \sigma^2\sum_{t=1}^{\Delta_2}\eta_{T_1+t}^2 \leq \frac{C_1}{\kappa}\mathbb{E}\left[\left(\mathbf{w}_{T_1+\Delta_2}^{(1)} - (\mathbf{w}^*)^{(1)}\right)^2\right] \leq C_1 f_{\text{lin}}(\mathbf{w}_{T_1+\Delta_2}) \leq \frac{C_1\tau\sigma^2}{T_1+\Delta_2}
$$

$$
\Rightarrow T_1 \leq (C_1\tau - 1)\Delta_2 \quad \Rightarrow \quad T_0 \leq \frac{(C_1\tau-1)^2}{C_1\tau}\Delta_2.
$$

Repeating this argument, we can show that

$$
T = T_0 \leq \frac{(C_1\tau-1)^i}{(C_1\tau)^{i-1}}\Delta_i \quad \text{and} \quad T_i \leq \frac{(C_1\tau-1)^{j-i}}{(C_1\tau)^{j-i-1}}\Delta_j \quad \forall\, i < j.
$$

We will use $i = 1$ in particular, which specializes to

$$
T_1 \leq \frac{(C_1\tau-1)^{j-1}}{(C_1\tau)^{j-2}}\Delta_j \quad \forall\, j \geq 2.
$$

Using the above inequality, we can lower bound the sum of $\Delta_j$ as

$$
\sum_{j=2}^{\kappa}\Delta_j \geq T_1 \cdot \sum_{j=2}^{\kappa}\frac{(C_1\tau)^{j-2}}{(C_1\tau-1)^{j-1}} \geq T_1 \cdot \frac{1}{C_1\tau} \cdot \sum_{j=2}^{\kappa}\left(1+\frac{1}{C_1\tau}\right)^{j-2}
$$

$$
\geq T_1 \cdot \frac{1}{C_1\tau} \cdot \exp\left(\kappa/(C_1\tau)\right). \tag{8}
$$

This means that

$$\mathbb{E}\left[f(\mathbf{w}_{T_i})\right] \geq \frac{d}{2} \cdot \mathbb{E}\left[\left(\mathbf{w}_{T_i}^{(2)} - (\mathbf{w}^*)^{(2)}\right)^2\right] \geq \exp(-6)\sigma^2 d \cdot \sum_{i=1}^{\Delta_1} \eta_{T+i}^2$$

$$\geq \frac{\exp(-6)\sigma^2 d}{\Delta_1} \geq \frac{\exp(-6)\sigma^2 d}{T_1} \geq \frac{\exp\left(\kappa/(C_1\tau) - 3\right)}{C_1\tau} \cdot \frac{\sigma^2 d}{\sum_{j=2}^{\kappa} \Delta_j},$$

where we used (8) in the last step. Rearranging, we obtain

$$\frac{\mathbb{E}\left[f(\mathbf{w}_{T_\kappa})\right]}{(\sigma^2 d/T_\kappa)} \geq \frac{\exp\left(\kappa/(C_1\tau) - 3\right)}{C_1\tau}.$$

If we choose a large enough $C$ (e.g., $3C_1$), the right hand side is at least $\frac{\exp((C/C_1)\log(\kappa+1) - 3)}{\kappa} \geq \kappa$. $\quad\square$

To prove Theorem 6, we rely on the following key lemma, which says if a query point $\mathbf{w}_T$ is bad (in the sense that it has expected value more than $10\tau\sigma^2 d/T$), then it takes at least $\Omega(T/\tau)$ steps to bring the error back down.

**Lemma 8.** *There exists universal constants $C_1, C_2 > 0$ such that for any $\tau \leq \frac{\kappa}{CC_1 \log(\kappa+1)}$ where $C$ is the constant in Theorem 5, suppose at step $T$, the query point $\mathbf{w}_T$ satisfies $f(\mathbf{w}_T) \geq C_1\tau\sigma^2 d/T$, then for all $\tilde{T} \in [T, (1 + \frac{1}{C_2\tau})T]$ we have $\mathbb{E}\left[f(\mathbf{w}_{\tilde{T}})\right] \geq \tau\sigma^2 d/T \geq \tau\sigma^2 d/\tilde{T}$.*

*Proof of Lemma 8.* Since $f(\mathbf{w}_T) \geq C_1\tau\sigma^2 d/T$ and $f(\mathbf{w}_T) = \frac{d}{2}\left(\kappa\left(\mathbf{w}_T^{(1)} - (\mathbf{w}^*)^{(1)}\right)^2 + \left(\mathbf{w}_T^{(2)} - (\mathbf{w}^*)^{(2)}\right)^2\right)$, we know either $\left(\mathbf{w}_T^{(1)} - (\mathbf{w}^*)^{(1)}\right)^2 \geq C_1\tau\sigma^2/2\kappa T$ or $\left(\mathbf{w}_T^{(2)} - (\mathbf{w}^*)^{(2)}\right)^2 \geq C_1\tau\sigma^2/2T$. Either way, we have a coordinate $i$ with eigenvalue $\lambda_i$ ($\kappa$ or 1) such that $\left(\mathbf{w}_T^{(i)} - (\mathbf{w}^*)^{(i)}\right)^2 \geq C_1\tau\sigma^2/(2T\lambda_i)$.

Similar as before, choose $\Delta$ to be the first point such that

$$\eta_{T+1} + \eta_{T+2} + \cdots + \eta_{T+\Delta} \in [1/\lambda_i, 3/\lambda_i].$$

First, by (6) or (7), we know for any $T \leq \tilde{T} \leq T + \Delta$, $\mathbb{E}\left[\left(\mathbf{w}_{\tilde{T}}^{(i)} - (\mathbf{w}^*)^{(i)}\right)^2\right] \geq \exp(-6)C_1\tau\sigma^2/(2\lambda_i T)$ just by the first term. When we choose $C_1$ to be large enough the contribution to function value by this direction alone is larger than $\tau\sigma^2/T$. Therefore every query in $[T, T + \Delta]$ is still bad.

We will consider two cases based on the value of $S^2 := \sum_{\tilde{T}=T+1}^{T+\Delta} \eta_{\tilde{T}}^2$.

If $S^2 \leq C_2\tau/(\lambda_i^2 T)$ (where $C_2$ is a large enough universal constant chosen later), then by Cauchy-Schwartz we know

$$S^2 \cdot \Delta \geq \left(\sum_{\tilde{T}=T+1}^{T+\Delta} \eta_{\tilde{T}}\right)^2 \geq 1/\lambda_i^2.$$

Therefore $\Delta \geq T/C_2\tau$, and we are done.

If $S^2 > C_2\tau/(\lambda_i^2 T)$, by Equation (6) and (7) we know

$$\mathbb{E}\left[\left(\mathbf{w}_{T+\Delta}^{(i)} - (\mathbf{w}^*)^{(i)}\right)^2\right] \geq \sigma^2 \sum_{\tilde{T}=T+1}^{T+\Delta} \eta_{\tilde{T}}^2 \exp\left(-2\lambda_i \sum_{j=\tilde{T}+1}^{T+\Delta} \eta_j\right)$$

$$\geq \exp(-6)\sigma^2 \sum_{\tilde{T}=T+1}^{T+\Delta} \eta_{\tilde{T}}^2 \geq \exp(-6) \cdot C_2\tau\sigma^2/(\lambda_i^2 T).$$

Here the first inequality just uses the second term in Equation (6) or (7), the second inequality is because $\sum_{j=\tilde{T}+1}^{T+\Delta} \eta_j \leq \sum_{j=T+1}^{T+\Delta} \eta_j \leq 3/\lambda_i$ and the last inequality is just based on the value of $S^2$. In this case as we can see as long as $C_2$ is large enough, $T + \Delta$ is also a point with $\mathbb{E}\left[f(\mathbf{w}_{T+\Delta})\right] \geq \lambda_i \mathbb{E}\left[\left(\mathbf{w}_{T+\Delta}^{(i)} - (\mathbf{w}^*)^{(i)}\right)^2\right] \geq C_1 \tau \sigma^2/(T + \Delta)$, so we can repeat the argument there. Eventually we either stop because we hit case 1: $S^2 \leq C_2\tau/\lambda_i^2 T$ or the case 2 $S^2 > C_2\tau/\lambda_i^2 T$ happened more than $T/C_2\tau$ times. In either case we know for any $\tilde{T} \in [T, (1 + 1/C_2)T]$ $\mathbb{E}\left[f(\mathbf{w}_{\tilde{T}})\right] \geq \tau\sigma^2/T \geq \tau\sigma^2/\tilde{T}$ as the lemma claimed. $\qquad\square$

Theorem 6 is an immediate corollary of Theorem 5 and Lemma 8.

*Proof of Theorem 7.* This result follows by running the constant and cut scheme for a fixed time horizon $T$ and then increasing the time horizon to $\kappa \cdot T$. The learning rate of the initial phase for the new $T' = \kappa \cdot T$ is $1/\mu T' = 1/\mu \cdot \kappa T = 1/LT$ which is the final learning rate for time horizon $T$. Theorem 2 will then directly imply the current theorem. $\qquad\square$

## D    DETAILS OF EXPERIMENTAL SETUP

### D.1    SYNTHETIC 2-D QUADRATIC EXPERIMENTS

As mentioned in the main paper, we consider three condition numbers namely $\kappa \in \{50, 100, 200\}$. We run all experiments for a total of $200\kappa$ iterations. The two eigenvalues of the Hessian are $\kappa$ and 1 respectively, and noise level is $\sigma^2 = 1$ and we average our results with two random seeds. All our grid search results are conducted on a $10 \times 10$ grid of learning rates $\times$ decay factor and whenever a best run lands at the edge of the grid, the grid is extended so that we have the best run in the interior of the gridsearch.

For the $O(1/t)$ learning rate, we search for decay parameter over $10-$points logarithmically spaced between $\{1/(100\kappa), 3000/\kappa\}$. The starting learning rate is searched over 10 points logarithmically spaced between $\{1/(20\kappa), 1000/\kappa\}$.

For the $O(1/\sqrt{(t)})$ learning rate, the decay parameter is searched over 10 logarithmically spaced points between $\{100/\kappa, 200000.0/\kappa\}$. The starting learning rate is searched between $\{0.01, 2\}$.

For the exponential learning rate schemes, the decay parameter is searched between $\{\exp(-2/N), \exp(-10^6/N)\}$. The learning rate is searched between $\{1/5000, 1/10\}$.

### D.2    NON-CONVEX EXPERIMENTS ON CIFAR-10 DATASET WITH A 44-LAYER RESIDUAL NET

As mentioned in the main paper, for all the experiments, we use the Nesterov's Accelerated gradient method (Nesterov, 1983) implemented in pytorch [12] with a momentum set to 0.9 and batchsize set to 128, total number of training epochs set to 100, $\ell_2$ regularization set to 0.0005.

With regards to learning rates, we consider $10-$values geometrically spaced as $\{1, 0.6, \cdots, 0.01\}$. To set the decay factor for any of the schemes such as 1,2, and 3, we use the following rule. Suppose we have a desired learning rate that we wish to use towards the end of the optimization (say, something that is 100 times lower than the starting learning rate, which is a reasonable estimate of what is typically employed in practice), this can be used to obtain a decay factor for the corresponding decay scheme. In our case, we found it advantageous to use an additively spaced grid for the learning rate $\gamma_t$, i.e., one which is searched over a range $\{0.0001, 0.0002, \cdots, 0.0009, 0.001, \cdots, 0.009\}$ at the $80^{th}$ epoch, and cap off the minimum possible learning rate to be used to be 0.0001 to ensure that there is progress made by the optimization routine. For any of the experiments that yield the best performing gridsearch parameter that falls at the edge of the grid, we extend the grid to ensure that the finally chosen hyperparameter lies in the interior of the grid. All our gridsearches are run such that we separate a tenth of the training dataset as a validation set and train on the remaining $9/10^{th}$ dataset. Once the best grid search parameter is chosen, we train on the entire training dataset and

---

[12]https://github.com/pytorch

evaluate on the test dataset and present the result of the final model (instead of choosing the best possible model found during the course of optimization).

