# OpenReview forum: "Rethinking learning rate schedules for stochastic optimization"
_ICLR.cc/2019/Conference_

### Official Review · AnonReviewer2 · 2018-10-31
**Interesting work. Can benefit from better experiments.**

**Rating:** 6
**Confidence:** 4

**Review:**

This work provides theoretical insights on recent learning rate proposals such as Cyclical Learning Rates (Smith et al.). The authors focus on stochastic approximation i.e. how large is the SGD loss as a function of condition number and horizon. The critical contribution is the theoretical benefit of oscillating learning rates over more traditional learning rate schemes. Authors provide novel upper/lower bounds to establish benefit of oscillating LR, support their theory with experiments and provide insights on finite horizon learning rate selection. An important drawback is that results only apply to linear regression which is a fairly simple setup.

I have two important comments regarding this work:
1) I believe proof of Theorem 3 has a bug. In the proof, authors use the inequality
(1-gamma_t lambda^k)^2 < exp(-2lambda^k gamma_t).
Obviously this can only be correct for gamma_t lambda^k<1. However, checking the setup of the problem, it can be seen that for largest eigenvalue and gamma_0, ignoring log factors:

gamma_0L = L/(mu T_e)=kappa / T_e=kappa/T.

Since, no restriction is imposed on T, gamma_0L can be as large as O(kappa) and invalidates the above inequality. So T should be T>O(kappa). I am not sure if this affects the overall statement or the remaining argument.

2) The paper can benefit from more detailed experiments (e.g. Figs 1 and 2). Arguably the most obvious baseline is "constant learning rate". However, authors compare to 1/T or 1/sqrt(T) learning rates. It is not at all clear from current experiments, if the proposed approach beats a good constant LR choice.

I am happy to increase my score if the comments above are addressed.

---

> ### Author Response · Authors · 2018-11-23
> **Author response**
>
> Thank you for your comments. Below we will present our response to your concerns:
>
> Regarding your first comment that T has to be O(kappa): you are absolutely right that this argument holds for T> O(kappa), which is the basic amount of time that the algorithm needs to be run to decay error (by a non-trivial amount) on the smallest eigen spaces. We will mention this condition on our theorem 3 explicitly (similar to the way the statement of theorem 2 has been written).
>
> Regarding the second comment about experimenting with constant learning rates for quadratics:
>
> We note that in the grid search parameter values for the 1/t decay scheme (in page 18), one end of the decay parameter makes the final learning rate only a factor of 3 smaller than the initial learning rate, which is reasonably close to a constant learning rate. The learning rate is also chosen to vary across a wide range of values starting from ones that certainly diverge to others that are super-tiny, resembling factors as small as 10/kappa^2 (page 18). Should the best performing hyper-parameter fall at the edge of a given grid search region, we have taken care to extend the grid search appropriately, indicating that if a constant step size performed best, we have extended our grid searches to obtain this specific case as well.
>
> Aside from this view, we note that we really wish to start our learning rate to be as close to the divergent point as possible. This is because, as one would realize, for learning rates as tiny as O(1/kappa^2)  (which is off the largest possible learning rate by a factor of 1/kappa), the burn-in (i.e. the rate of bias decay) may actually be pretty bad, i.e., off by an extra condition number.
>
> Finally, regarding "simplicity" of least squares:
>
> Our observation in this paper is that highly subtle issues such as non-convergent step size scheme's advantages over traditional polynomial decay schemes show up even when solving a strongly convex quadratic problem, where, there is scope to perform exact computations to analyze the progress of the algorithm, and while providing lower bounds on standard poly decay step size strategies as well as on the behavior with an infinite horizon. Note that upper bounds on behavior of algorithms that can be interpreted as employing exponentially decaying step sizes for smooth convex/non-convex optimization (in terms of gradient norm) have been presented in Allen-Zhu (2018) (as mentioned in the paper). As for lower bounds, note that a lower bound for a specific problem instance spells bad news, for e.g., lower bounds for (smooth+convex) deterministic optimization have been proven with a quadratic objective.

---

### Official Review · AnonReviewer1 · 2018-11-02
**neat technical results, but misleading narrative**

**Rating:** 4
**Confidence:** 4

**Review:**

The paper studies the effect of learning-rate choices for stochastic optimization, focusing on least-mean-squares with decaying stepsizes. The main result is showing that exponentially decaying stepsizes can yield improved rates of convergence of the final iterate in terms of dependence on the condition number. The proposed learning rate schedule depends on the condition number and the number of iterations. This positive result is complemented by showing that without prior knowledge of the time horizon, any stepsize sequence will frequently yield suboptimal solutions.

I have mixed feelings about the paper. On the positive side, the particular observation that exponential learning-rate schedules lead to faster convergence for SGD in linear least-squares problems indeed seems to be a novel result, and the lower bound also appears to be new and interesting. The analysis seems to be technically correct as well.

On the other hand, I have several concerns about the presentation of the results:

- The abstract and the introduction sets up a misleading narrative around the results: the authors seem to suggest that their work somehow explains why certain learning-rate schedules work better than others for deep learning applications / non-convex optimization, although the actual results exclusively concern the classical problem of linear least-squares regression. This presentation is completely uncalled for as the authors themselves admit that it is unclear how the results would generalize to other convex optimization settings, let alone non-convex optimization. Also, I think that this presentation style is rather harmful as it suggests that learning-theory results concerning classical setups are somehow embarrassing, so they need to be sold through some made-up connections to trendy topics in deep learning. I would suggest that the authors completely "rethink" the presentation of the paper and write it in a style that is consistent with the actual results: as a learning theory paper, without the irrelevant deep learning experiments (that only show well-known phenomena anyway).

- The paper misrepresents a large body of work on stochastic/online optimization. Specifically, the authors suggest that the stochastic optimization literature exclusively suggests the use of polynomially decaying stepsizes. This picture is grossly inaccurate for multiple reasons:
*** It has been known for a while that the de facto optimal tuning of SGD for least squares involves a large constant stepsize and iterate averaging (see, e.g., Bach and Moulines, NIPS 2013). This approach is only mentioned in passing without any discussion, even though it yields convergence rates that do not involve *any* dependence on the condition number in the leading term---thus achieving a much more significant improvement than the learning-rate schedule studied in this paper. In light of these results, learning-rate schedules are already being "re-thought" as we speak, and studying the behavior of the last iterate has received less attention in the past couple of years. If anything, the present paper only provides further evidence (through the negative result) that the individual iterates are ill-behaved in general and it is better to average the iterates instead. I would consider this negative result as an interesting addition to the stochastic-optimization literature, had it been presented in a completely different narrative (e.g., augmenting the discussion in "Stochastic gradient descent for non-smooth optimization: Convergence results and optimal averaging schemes" by Shamir and Zhang, 2013).
*** Exponentially decaying (or "constant-and-cut", as they are called here) schedules have actually been studied before in the paper "Beyond the Regret Minimization Barrier: Optimal Algorithms for Stochastic Strongly-Convex Optimization" by Hazan and Kale (JMLR 2014). This significantly weakens the main intended selling point of the paper which was being the first-ever study of such learning-rate schedules. The results in said paper are of a somewhat different nature, but they have arguably as little to do with deep learning as the results of the present paper has. Notably, both the present paper and the cited work rely on *strong convexity* of the objective (through assuming prior knowledge of the condition number), so I would expect that none of these results would explain anything in the context of deep learning.

On the technical side, the proofs appear to be correct but presented somewhat sloppily, with most of the notation appearing without proper definitions. For instance, the proof of Theorem 2 seems to import notation from the proof of Theorem 1, although without explicitly mentioning that the covariance matrix is assumed to be diagonal(ized). The proof of Theorem 3 then seems to again replace this previously (non-)established notation by another one (e.g., v becomes err and \eta becomes \gamma). The proofs also involve long sequences of inequalities without explanation, and only bound the variances (w_k-w^*_k)^2 without mentioning how this quantity is related to the excess risk. (The relation is well-known but not obvious at all for first-time readers of such proofs.)

One technical limitation of the results is that they assume a simple additive-noise model for the gradients, which the authors conveniently call "fairly natural" and incorrectly claim to hold for linear regression with well-specified models (footnote 8). In reality, the gradient noise in this setting also depends on the current iterate w_t, which makes analysis significantly harder. (To see the difference, just compare the complexity of the proofs of Lemma 1 and Theorem 2 that correspond to these different settings in "Harder, Better, Faster, Stronger Convergence Rates for Least-Squares Regression" by Dieleveut, Flammarion and Bach, 2017.)

Overall, I don't think that this paper is fit for publication in its present form. Once again, I would suggest that in a future version, the authors focus solely on discussing the actual results without attempting to draw disproportionate conclusions from them.

Detailed comments
=================
- pp.1, abstract: the first half of the abstract is completely irrelevant to the rest of the paper, so I'd suggest removing it.
- pp.1, "learning-rate schedules for SGD is a rather enigmatic topic"---"enigmatic" feels like a bit of a strong adjective here, given that there are many aspects of learning-rate tuning that are actually pretty well-understood.
- pp.2: The second paragraph on page 2 is again irrelevant to the actual technical content of the paper.
- pp.2, "all the works in stochastic approximation try to bound the error of each iterate of SGD"---This is simply not true, given the growing literature concerning the behavior of the *averaged iterates*.
- pp.4, first display: poor typesetting.
- pp.6, Eqs. 1--3: ditto.
- pp.8, last paragraph: Singling out the particular setting of gradient-norm minimization feels arbitrary and poorly justified.
- pp.11: the first and second displays should be switched for better readability (otherwise the first one comes without explanation). Also note that this form is not just due to the algorithm design, but also to the simplified noise model.
- pp.12, App B:
*** It appears that you forgot to mention here that you're working in the coordinate system induced by the eigenvectors, and also forgot to define the eigenvalues, etc.
*** The indices (1) and (k) are incoherent in the first display.
*** Although you promise you'll prove the inequality in the second display, you eventually prove something else.
*** It is not very clear on first sight that \ell^* actually exists and falls within the scope of \ell---you should explain that it exists due to the choice of the number of phases. (Which, by the way, should be rounded up to allow this property?)
*** The sequence of inequalities in the last display seems correct but unnecessarily hard to verify due to the lack of explanations.

---

> ### Author Response · Authors · 2018-11-23
> **Author response**
>
> Part 1/2:
>
> Thank you for your feedback. Below, we will present some clarifications to your points:
>
> --- With regards to "mis-leading narrative for abstract/introduction":
>
> As mentioned in the paper, Allen-Zhu(2018) presents strong upper bounds (for gradient norm) for schemes that can be interpreted as exponentially decaying step sizes for smooth convex and non-convex optimization. However it does not deal with any lower bounds, and in particular, it does not show a gap between polynomial and exponential step size schemes, or about the issues arising in the infinite horizon setting.
>
> Our paper performs a tight analysis for a strongly convex quadratic and shows a gap between exponential decaying schemes and other convergent polynomial decay schemes for the fixed horizon case, and also a negative result for the infinite horizon case. Furthermore, with regards to lower bounds, issues highlighted with a specific problem (like optimizing a quadratic) spells bad news, as is the case with deterministic smooth convex optimization, where the lower bounds on rates for first-order methods are obtained by providing a quadratic objective.
>
> Empirically we show this difference does manifest when training neural networks. To the best of our knowledge, there are no papers that show the performance of polynomial decay schemes with exhaustive grid searches for training some of these near state-of-the-art neural nets. Moreover, note that continuous exponentially decaying step size schemes (as used in the simulations) is much easier to tune (with one lesser hyper parameter) compared to constant and cut schemes -- we advocate using continuous exponential decay scheme compared to existing schemes that practitioners employ for training their state-of-the-art deep learning models. As a final remark, note that some of these benefits (in practice) are highlighted in recent works in the deep learning literature, ex, refer to Loshchilov & Hutter, 2016 reference in the paper.
>
> We really believe that this interpretation along with capturing gaps between classical stochastic approximation theory and modern SGD implementations for practical deep learning based problems presents a coherent argument for why this narrative is right headed.
>
>
> --- Regarding "misrepresentation of large body of work in stochastic/online optimization":
>
> -- With regards to the recent series of efforts including Moulines and Bach (2013), Defossez and Bach (2015) etc., that rely on constant step size + iterate averaging:
>
> We do note that iterate averaging (+ constant step sizes) offer minimax rates (in page 4) and will make it more explicit.
>
> The reason this paper is all about the final iterate (i.e. without iterate averaging):
>
> Nearly all practical SGD implementations employ constant and cut schemes + final iterate (i.e. no iterate averaging) to obtain close to the best behavior for a specific computation time (say, 100 passes over a dataset). Stochastic Approximation theory papers show anytime optimal results with polynomially decaying schemes + iterate averaging, which is in stark contrast to SGD as used in practice. However, these theory results appear lacking in terms of explaining why practitioners prefer drastically different prescriptions for implementing SGD. Our paper begins reconciliation of the theory side to account for what is often mentioned as the "art" of tuning step sizes for deep learning problems. There is in fact a strong connection that our paper highlights, which, in our view, empirical science has already uncovered.
>
>
>
> -- With regards to the effort of Hazan and Kale (2014): Thanks for this pointer and we will cite this paper. This is for the following reasons:
>
> - Note that despite the fact that Hazan and Kale (2014) use constant and cut schemes, the duration for which they keep holding the learning rate to be a constant is actually *doubling* every time they cut the learning rate, implying that this is not an exponentially decaying step size scheme with a "constant" decay factor (which is a distinct theme of both practical implementations of SGD as well as our paper).
>
> - Hazan and Kale (2014) serve to improve certain log factors in their convergence rates (improve logT/T rate to 1/T rate, which of course is worthwhile), whereas, our paper presents *exponential* improvements in the variance term (Kappa/T to log Kappa/T). We hope that the reviewer agrees that this certainly is a tremendous improvement.
>
> Owing to the above differences, as well as other points discussed above, we believe that our paper presents a strong message with regards to phenomenon observed in practice when training the current generation of deep learning models.

---

> > ### Author Response · Authors · 2018-11-23
> > **Author response (part 2/2)**
> >
> > Part 2/2
> > --- Differences in Oracle models: We have already mentioned this distinction in the paper and did mention that the other oracle model (considered, say in Bach and Moulines (2013)) is well worth considering as a part of future work (in page 4). We will remove the statement the noise model is fairly natural. You are perfectly right that with tiny batch sizes these oracles are indeed different and we do mention this in related work and point to the work of Jain et al. (2017) which highlights these distinctions. Our intention here was that with large minibatch sizes, both the models are fairly equivalent. Note that our work presents results that suggests rethinking of an area that has been around for six to seven decades and banks on a "textbook" assumption (say, Bubeck(2014) and others) that has been considered in its seminal effort [Robbins and Monro, 1951] and several recent works (Lacoste-Julien (2012), Rakhlin et al (2012), Shamir and Zhang (2013), Allen-Zhu (2018)).
> >
> > Regarding proof presentation: We will work on improving the presentation of our proofs and incorporating your detailed comments. Thank you very much for the same.
> >
> >
> > We hope that the reviewer considers why these results (both upper and lower bounds) are foundational to reconsider algorithm design for practical Machine Learning models (including the modern neural net models) and reconsiders her/his decision that this paper is already rejected.

---

> > > ### Comment · AnonReviewer1 · 2018-12-02
> > > **response**
> > >
> > > Thanks for the thoughtful response. I apologize for my own misrepresentation of the result of Hazan and Kale (2014)---I withdraw my original criticism regarding the lack of the discussion of this paper.
> > >
> > > That said, I still stand by my view that the presentation of the results in the paper is inappropriate and quite misleading, and that the paper would benefit better from a more honest presentation as a theory paper. To an extent, it is true that the results justify the use of exponentially decaying stepsizes, but only do so in a restricted setting while heavily relying on strong assumptions (strong convexity, smoothness). It's an extremely long stretch to claim that this would justify such stepsizes in deep learning applications.
> > >
> > > I strongly encourage the authors to resubmit their work after working out this presentation issue.

---

> > > > ### Author Response · Authors · 2018-12-02
> > > > **response**
> > > >
> > > > Dear reviewer,
> > > >
> > > > Thank you for your response.
> > > >
> > > > We would like you to take a look at the work of Allen-Zhu (2018), which presents two loop procedures that parallel the result of our paper for general smooth convex/non-convex optimization. Allen-Zhu's schemes can precisely be interpreted as exponentially decaying stepsizes. It is worthwhile to rethink these schemes (if you may) to develop a single loop procedure that obtains a result similar to the spirit of this paper.
> > > >
> > > > We strongly emphasize that these observations are *not* just "theory statements"; our empirical results prove these claims for a realistic deep learning problem (in practice); paralleling views of Allen-Zhu (2018)'s theory results and our own upper/lower bounds. We present our finding that a continuous exponential decay scheme helps to prevent babysitting learning rates that are considered the norm by practitioners (see courses that mention this fact: http://cs231n.github.io/neural-networks-3/#baby). We precisely feel that this paper belongs in this conference where there exists a sizeable set of practitioners who will benefit from this finding.
> > > >
> > > > We hope the reviewer reconsiders her/his views about this paper given the results and its connections that presents the *first* progress towards demystifying the apparent "art of stepsize tuning".

---

### Official Review · AnonReviewer4 · 2018-11-13
**Well written, some parts require clarification**

**Rating:** 6
**Confidence:** 4

**Review:**

This paper presents a theoretical study of different learning rate schedules. Its main result are statistical minimax lower bounds for both polynomial and constant-and-cut schemes.

I enjoyed reading the paper and I think the contributions in it shed some light in step size schedules that have shown to be useful in practice. I do have however some concerns that I hope the authors can address in their rebuttal. My initial rating is marginally below acceptance but I will gladly increase this rating if my concerns are addressed.


# Pros

* The paper is written in a way that's both clear and accessible.

* The Theoretical contributions are important, as they address the choice of step size in one of the most used optimization methods machine learning and are novel to the best of my knowledge.

* Due to time constraints, I only skimmed through the proofs, but results seem correct.


# Concerns


My biggest concern is that its unclear how realistic is their noise model. The authors assume that the noise in the stochastic gradients e verifies E[e e^T ] = \sigma H. While they claim that this is verified for problems like least squares, it is not clear to me that this is indeed the case. Related work like (Moulines and Bach, 2013) and (Flammarion and Bach, 2015) take the same setting but can only assume that the covariance of the noise is _bounded_ by a matrix of sigma times H. How do the authors obtain a much stronger condition on the noise covariance with the same assumption? I would be much more convinced with a proof in appendix clearly showing that the assumptions in footnote 8 imply the aforementioned covariance of the noise and a paragraph comparing their noise model with that of related literature like the aforementioned references (I'm not affiliated with any of that work).

Also, the authors claim that their results hold for an arbitrary noise covariance matrix but the proofs are all done with the specific \sigma H matrix. I don't think its OK to say "our results hold for a more general setting" without proof. If they do hold for a more general setting then the proofs should be done in the general setting. If not, it should only be mentioned as future work. Please edit that remark accordingly.

* The paper does not compare or discuss against constant step size with averaging, which has been shown to be theoretically optimal in some scenarios (see aforementioned papers). This should at least be mentioned, and ideally also included in experiments.


# Presentation issues

Clarity of the proofs can be improved. For example, in Theorem 1, the formula for v_T(1) and v_T(d) follow from a recurrence that is stated _below_ the formula, needing several passes to understand. The proofs could benefit from a pass on them to improve the flow.


It is never clear whether expectations are taken with respect to the full randomness of the algorithm or conditioned on previous randomness. The E in Eq. just-before-section-4 (please add equation numbers) is a full expectation while the E[e] should be conditioned on previous randomness. The expectation in footnote 8 is also unclear if its wrt to the stochasticity of the algorithm or the randomness in the data generating process.


No equation numbers  makes it difficult to reference equations. Please add equation numbers so that reviewing is not more difficult than it should (and others can reference your work more precisely).

Other minor presentation issues include:

  * Page 1: Why l-BFGS and not L-BFGS? the lowercase l makes it look like a 1.
  * Page 2: There important -> There *are* important.
  * Page 2: In fact, at least ... (missing parenthesis around Omega tilde).
  * Page 4: "a stochastic gradient oracle which gives us" the second w should also be boldface.
  * Page 4: I would have appreciated

  * Page 11: "variance in the i-th" direction. It would be more correct to say in the i-th coordinate as otherwise it can be mistaken with the i-th update direction.

Update:
  I am satisfied with the answers and have upgraded my rating.

---

> ### Author Response · Authors · 2018-11-23
> **Author response**
>
> Thank you for your feedback. Below, we will address your concerns:
>
> Issues with noise model:
>
> ***Response:***
> Regarding noise model: While the assumption is written as E[e e^T]=\sigma^2 H, our upper bound results (and proofs) hold true even when E[e e^T]\preceq \sigma^2 H (where \preceq refers to the PSD ordering between a pair of matrices), in the sense of how Moulines and Bach (2013), Flammarion and Bach (2015) work. We can change this assumption and proofs in the paper to reflect accordingly. As for the lower bounds, the goal is to come up with a problem instance which suffers a specific error, so the assumption E[e e^T]=\sigma^2 H suffices.
>
> Noting that our argument applies to the situation when E[e e^T]\preceq\sigma^2 H, our results apply to the general covariance matrix case with \sigma being the smallest number satisfying V \preceq \sigma^2 H, and we will mention this explicitly to avoid any confusion.
>
> Finally, we note the differences in oracle models in the final paragraph in the related works in theory section (page 3). Please refer to Jain et al (2017) for more information regarding distinctions in the oracle models and future work in this regard.
> -------------------------------------------------------------------------------------------------------
>
> Constant step size + averaging:
>
> ***Response:***
> In practice, what is used predominantly is the final iterate (no iterate averaging) + exponential decaying stepsize, whereas, theory papers suggest averaged iterate + polynomial decay stepsize (or constant stepsize+iterate averaging for least squares).  We do mention that constant stepsize + averaging achieves optimal rate (in an any time sense) in page 4 (and we will make it more explicit), but this paper's message is all about the behavior of the *final* iterate, since this is of real interest for nearly all practical SGD implementations.
> -------------------------------------------------------------------------------------------------------
>
> Presentation issues:
>
> ***Response:***
> We will present a complete derivation with regards to the variance in various directions, clarify expectations (and conditional expectations) and add in equation numbers and address your minor issues as well. Thanks for pointing these out.

---

> > ### Comment · AnonReviewer4 · 2018-12-04
> > **Satisifed**
> >
> > Thanks for the answers, I have upgraded my rating accordingly.

---

### Meta-Review · Area_Chair1 · 2018-12-18
**Borderline paper**

**Confidence:** 3
**Recommendation:** Reject

**Metareview:**

R4 recommends acceptance while R2 is lukewarm and R1 argues for rejection to revise the presentation of the paper. As we unfortunately need to reject borderline papers given the space constraints, the AC recommends "revise and resubmit".